# The N-Linked Glycosylation Site N201 in eel Lutropin/Choriogonadotropin Receptor Is Uniquely Indispensable for cAMP Responsiveness and Receptor Surface Loss, but Not pERK1/2 Activity

**DOI:** 10.3390/cimb47050345

**Published:** 2025-05-09

**Authors:** Munkhzaya Byambaragchaa, Dong-Wan Kim, Sei Hyen Park, Myung-Hwa Kang, Kwan-Sik Min

**Affiliations:** 1Carbon-Neutral Resources Research Center, Hankyong National University, Anseong 17579, Republic of Korea; munkhzaya_b@yahoo.com; 2Genetic Engineering, Hankyong National University, Anseong 17579, Republic of Korea; 3Graduate School of Animal BioScience, Hankyong National University, Anseong 17579, Republic of Korea; iwoorimil@daum.net (D.-W.K.); mrtree119@naver.com (S.H.P.); 4Department of Food Science and Nutrition, Hoseo University, Asan 31499, Republic of Korea; mhkang@hoseo.edu; 5Division of Animal BioScience, School of Animal Life Convergence Sciences, Hankyong National University, Anseong 17579, Republic of Korea

**Keywords:** eel LH/CGR, N-linked glycosylation, cAMP responsive, pERK1/2 activity

## Abstract

The seven transmembrane-spanning lutropin/chorionic gonadotropin receptors (LH/CGRs) trigger extracellular signal-related kinases (ERK1/2) via a noticeable network dependent on either G protein (Gαs) or β-arrestins. LH/CGRs are highly conserved, with the largest region within the transmembrane helices and common N-glycosylation sites in the extracellular domain. We aimed to determine the glycosylation sites that play crucial roles in cAMP and pERK1/2 regulation by constructing four mutants (N49Q, N201Q, N306Q, and N312Q). The cAMP response in cells expressing the N201Q mutant was completely impaired, despite high-dose agonist treatment. The cell-surface expression level was lowest in transiently transfected cells, but normal surface loss of the receptor occurred in cells expressing the wild-type and other mutant proteins. However, the N201Q mutant was only slightly reduced after 5 min of agonist stimulation. All mutants showed a peak in cAMP signaling 5 min after stimulation with a pERK1/2 agonist. Of note, cAMP activity was completely impaired in the N201Q mutant; however, this mutant still displayed a pERK1/2 response. These data show that the specific N-linked glycosylation site in eel LH/CGR is clearly distinguished by its differential responsiveness to cAMP signaling and pERK1/2 activity. Thus, we suggest that the cAMP and pERK1/2 signaling pathways involving eel LH/CGRs represent pleiotropic signal transduction induced by agonist treatment.

## 1. Introduction

The lutropin/choriogonadotropin receptor (LH/CGR), along with follicle-stimulating hormone receptor (FSHR) and thyroid-stimulating hormone receptor (TSHR), is a member of the glycoprotein hormone receptor family that belongs to class A (rhodopsin-like) G-protein-coupled receptors (GPCRs). These receptors contain a typical seven-transmembrane domain and a large extracellular N-terminal domain responsible for ligand binding [1]. LH/CGR is primarily expressed in the ovaries and testes and is known to regulate the early phase of gametocyte formation, follicle maturation, and ovulation by binding to LH or CG [2]. LH/CGR is predominantly expressed in Leydig cells, where it regulates testosterone production and indirectly supports spermatogenesis. In the ovaries, it is mainly detected in theca cells in preovulatory follicles, granulosa cells in mature follicles, and the corpus luteum [3]. These GPCRs couple with Gαs protein to activate adenylyl cyclase and increase cAMP levels. They also activate β-arrestin-meditated pathways and phosphorylated extracellular signal-regulated kinase 1/2 (pERK1/2) signaling [4,5].

All GPCRs undergo at least one modification during their lifetime, either co-translationally during biosynthesis or post-translationally after synthesis and delivery to the cell surface [6]. Post-translational modifications of GPCRs are completed through various important processes, including phosphorylation, SUMOylation, ubiquitination of intercellular regions, N- and O-linked glycosylation of extracellular domains, and palmitoylation. Glycosylation regulates post-translational modifications of GPCRs, including receptor folding, trafficking, ligand binding, signaling, and dimerization. N-glycosylation is one of the most common modifications of GPCRs. Thus, the well-known type of N-linked glycosylation occurs at approximately 70% of consensus sequences [7]. Similar to other GPCRs, LH/CGRs and FSHRs contain several conserved N-linked glycosylation sites in their extracellular domains. The functional roles of N-linked glycosylation of glycoprotein hormone receptors in signal transduction, folding, receptor binding, and cell-surface expression have been reported in multiple studies [8,9,10,11,12,13].

In rat LH/CGR, N-linked oligosaccharides of mature receptors are not required for hormone binding, as demonstrated by glycosidase treatment or the inhibition of glycosylation using tunicamycin in Leydig tumor cells and rat granulosa cells [2]. Another study suggested that N-linked glycosylation is not required for the proper folding of LH/CGR into a mature receptor capable of binding hormones and participating in signaling [12]. The N-linked glycosylation site in rat LH/CGR, Asn^173^, is crucial for proper receptor trafficking because its absence results in significantly decreased activity and prevents expression on the plasma membrane, demonstrating that Asn^173^ is necessary for proper receptor localization [8]. Although N-linked glycosylation in rat FSHR is not required for ligand binding, two sites (Asn^174^ and Asn^276^) are glycosylated. A single mutation allows expression and ligand binding with normal affinity, whereas double mutations prevent efficient folding necessary for high-affinity binding [11]. Two glycosylation sites at Asn^77^ and Asn^113^ of human TSHR are necessary for the expression of an active receptor on the cell surface. However, glycosylation at Asn^99^, Asn^177^, Asn^198^, and Asn^302^ does not affect the cAMP response or ligand binding [14].

Recently, we focused on the functional roles of GPCR signal transduction in activating and inactivating LH/CGR and FSHRs, demonstrating that these highly conserved regions in glycoprotein hormone receptors show almost identical functionality in mammals and fish [15,16,17]. GPCRs are versatile molecules that adopt alternate spatial arrangements. Accordingly, different ligands either activate specific conformations of the same GPCR or induce the activation of distinct signaling pathways, such as G protein or β-arrestin pathways [18]. Thus, one of the functions of N-glycosylation in GPCRs is described as contributing to biased signaling and functional selectivity.

There have been no reports of the comparative analysis of G protein signaling and pERK1/2 in relation to the N-linked glycosylation sites of eel LH/CGR. In this report, we illustrate the effect of mutations at potential glycosylation sites (Asn^49^, Asn^201^, Asn^306^, and Asn^312^) on receptor expression, surface loss, and biological activities of cAMP and pEKR1/2. Our results clearly revealed that N-glycosylation at Asn^201^ is absolutely necessary for the cAMP signal response, expression, and surface loss of the receptor, but not for pERK1/2 activity.

## 2. Materials and Methods

### 2.1. Construction of Glycosylation Mutants of eel LH/CGR cDNA

To construct each mutant, overlapping polymerase chain reaction (PCR) was performed using the full-length cDNA of eel *LH/CGR*, which was previously cloned in our laboratory [17]. Each glycosylation site was modified by replacing Asn (AAC) with Gln (CAA). Four glycosylation sites at amino acid positions 49, 201, 306, and 312 were predicted to be N-linked, and mutants were constructed for these sites. Briefly, two different sets of PCRs were conducted. PCR amplification was performed using a wild-type-sequence-specific forward primer containing an XhoI restriction enzyme site and a reverse primer containing the mutation site. Subsequently, the downstream region was amplified using a forward primer specific to the mutation site and a reverse primer specific for eel *LH/CGR*, which included an EcoRI restriction site. After each fragment was amplified, a second round of PCR was performed using both fragments simultaneously as templates. The full-length sequence was amplified using wild-type-sequence-specific forward and reverse primers subcloned into the pGEM-T easy vector (Promega, Madison, WI, USA) and sequenced to check for PCR errors. Finally, the mutant eel LH/CGRs were subcloned into the expression vector pCORON1000 SP VSV-G and pcDNA3.1 (Amersham Biosciences, Piscataway, NJ, USA). The receptor cDNA cloned into the pGEM-T Easy vector was digested with XhoI and EcoRI restriction enzymes, whose recognition sites were introduced by PCR primers. The digested fragments were then inserted into the corresponding sites of the pVSV-G expression vector. The full-length eel LH/CGR cDNA, including the signal sequence, was also digested with EcoRI and XhoI and subcloned into the same sites of the pcDNA3 mammalian expression vector. In the pCORON1000 SP-VSV-G vector, the receptor signal sequence was excluded and the VSVG region from the expression vector was utilized. The plasmids were designated eel LH/CGR-N49Q, -N201Q, -N306Q, and -N312Q, respectively. A representation of the mutation sites in eel *LH/CGR* is shown in Figure 1.

### 2.2. Transient Transfection into CHO-K1 and HEK 293 Cells

CHO-K1 cells were cultured in growth medium comprising Ham’s F-12 medium supplemented with 2 mM glutamine, 50 U/mL penicillin, 50 μg/mL streptomycin, and 10% fetal bovine serum (FBS). HEK 293 cells were cultured in Dulbecco’s modified Eagle medium (DMEM) supplemented with 10 mM HEPES, 50 μg/mL gentamicin, and 10% FBS. One day before transfection, the cells were seeded in a six-well plate. Transfection was performed the next day or when the cells reached approximately 80–90% confluence. Each eel LH/CGR mutant plasmid was diluted with the transfection reagent Lipofectamine 2000 and incubated at room temperature for 10–15 min. After washing the cell culture medium twice, the transfection reagent and plasmid mixture were added dropwise to the cells. After 5 h, the medium was supplemented with an equal volume of medium containing 20% FBS to obtain a final concentration of 10% FBS.

### 2.3. cAMP Analysis Using Homogeneous Time-Revolved Fluorescence

The cAMP Dynamic 2 Homogeneous Time-Revolved Fluorescence (HTRF) assay was performed following the supplier’s protocol (Cisbio, Codolet, France) and was based on a method previously used in our laboratory [16]. A brief description is provided below. MIX was added to prevent cAMP degradation in the culture medium. A standard cAMP concentration range of 0.17–712 nM was used. The rec-eel LH ligand was then added, and the cells were incubated for 30 min. cAMP-d2 and anti-cAMP-cryptate were added, and the cells were incubated at room temperature for 1 h. After the reaction, fluorescence was measured at 665 and 620 nm using a TriStar^2^ LB942 microplate reader (BERTHOLD Tech., Wildbad, Germany). The results are expressed as delta F% (665 nm/620 nm ratio) to reflect cAMP inhibition, and the cAMP concentration was determined by comparison with cAMP concentration standards using GraphPad Prism 6.0 (GraphPad, San Diego, CA, USA).

### 2.4. Agonist-Induced Loss of Cell-Surface Expression of Receptors

The expression and surface loss of eel LH/CGR were assessed using enzyme-linked immunosorbent assays (ELISAs) as previously described [17]. Briefly, cells were seeded into 96-well plates at 24 h post-transfection. The following day, the cells were preincubated with rec-eel LH (250 ng/mL) for different times (0, 5, 15, and 30 min). The cells were then washed with Dulbecco’s phosphate-buffered saline (PBS) and fixed with 4% paraformaldehyde for 5 min. After blocking for 30 min, the cells were incubated with rabbit anti-VSVG antibody (1:1000) and horseradish-peroxidase-conjugated anti-rabbit antibody (1:4000) (Cell Signaling Technology, Danvers, MA, USA) for 1 h. Finally, ELISA Femto Maximum Substrate and Luminol/Enhancer were added, and the luminescence signal was measured using a Cytation 3 plate reader. The cell-surface expression levels of eel wild-type (LH/CGR-wt) and mutant LH/CGR were set to 100% in the untreated cells. Surface loss of the receptor was calculated by comparing the loss induced by agonist stimulation with the level in untreated cells (defined as 0% surface receptor loss).

### 2.5. Measurement of pERK1/2 Activity Using HTRF

For the pERK1/2 experiment using the HTRF analysis system, HEK-293 cells were seeded in a 96-well plate coated with poly-L-lysine for 10 min 24 h after transfection. The following day, rec-eel LH (250 ng/mL) was added, and the cells were incubated for 0, 5, 10, 15, and 30 min. First, the wells were washed once with PBS, after which 50 μL of lysis buffer was added to each well and the plate was shaken at room temperature to lyse the cells. Then, 16 μL of the lysate was transferred to a 384-well plate, followed by the addition of pERK1/2 Eu Cryptate and an anti- pERK1/2 d2 antibody. The fluorescence of total ERK and pERK1/2 was measured at 620 and 665 nm after 4–24 h of incubation. The pERK1/2 values are presented as actual measured values for comparison, and the results are expressed as fold changes by normalizing each time-zero value to 1. Additionally, the time-zero value of the wild-type receptor was set to 1 for comparative analysis.

### 2.6. Analysis of pERK1/2 Activation by Western Blotting

At 48 h post-transfection, HEK-293 cells were starved for at least 6 h and then stimulated with rec-eel LH (250 ng/mL) for different times. The cells were lysed with radioimmunoprecipitation assay buffer (Sigma-Aldrich, St. Louis, MO, USA). Equal amounts (5–10 μg) of cellular extracts were electrophoresed through 12% sodium dodecyl sulfate-polyacrylamide gels and transferred onto polyvinylidene difluoride membranes. pERK1/2 and total ERK1/2 were detected via immunoblotting using rabbit polyclonal anti-p44/42 MAPK (1:2000) and anti-MAPK1/2 antibodies (1:3000) (Cell Signaling Technology, Danvers, MA, USA), respectively, with overnight incubation at 4 °C. The membranes were then incubated with an anti-rabbit secondary antibody for 1 h. Chemiluminescence was detected using SuperSignal^TM^ West Femto Maximum Sensitivity Substrate (Thermo Fisher Scientific, Waltham, MA, USA), and pERK1/2 immunoblots were quantified by densitometry using Image Lab v6.0 (Bio-Rad, Hercules, CA, USA).

### 2.7. Data Analysis

DNA sequence analysis was performed using the Multalin multiple sequence alignment tool. GraphPad Prism 6.0 (San Diego, CA, USA) was used to analyze cAMP concentration data, including delta F% raw data and EC_50_ values, and to construct cAMP graphs. The GraFit 5 program (Erithacus Software, Surrey, Horley, UK) was used to generate pERK1/2 graphs displaying percentage ratios and fold values. Statistical analysis was performed using one-way analysis of variance (ANOVA), followed by Tukey’s multiple comparisons test using GraphPad Prism software (version 6.0).

## 3. Results

### 3.1. Expression at the Cell Surface After Mutation of Putative N-Linked Glycosylation Sites

As shown in Figure 1, four putative N-linked glycosylation sites were identified in the extracellular domain of eel LH/CGR. One mutant (N201) was highly conserved among the LH/CGRs of mammalian species and eels. This site was confirmed to be located in exon 7 of *LH/CGR*. Specifically, the N49 site of eel LH/CGR does not exist in other species, except equine LH/CGR. The other two sites (N306 and N312) were located at nearly the same positions as in the LH/CGR of mammals, based on an amino acid comparison analysis.

To determine whether glycosylation sites play a significant role in the signal transduction pathway of eel LH/CGR, we constructed eel LH/CGR mutants using overlapping PCR, in which the potential glycosylation sequence was changed by substituting the Asn (N) residue with Gln (Q). The expression results, as shown in Figure 2, confirmed that eel LH/CGR-wt and the N312Q mutants were efficiently expressed in HEK 293 cells. However, three other mutants (N49Q, N201Q, and N306Q) exhibited considerably lower expression levels. Two mutants (N49Q and N306Q) exhibited a significant decrease in expression levels to approximately 68% and 65% of eel LHR-wt levels, respectively. The expression level of the N201Q mutant was only 40% of that of LH/CGR-wt, which was the lowest among the mutant proteins, despite being a highly conserved N-linked glycosylation site in both mammals and eels. Therefore, the N-linked glycosylation site, N201Q, in eel LH/CGR has an extraordinary influence on cell-surface expression.

### 3.2. Biological Activities of eel LH/CGR-Wt and N-Linked Glycosylation Mutants

The effects of the mutant of the glycosylation sites on the rec-eel LH–stimulated cAMP response are summarized in Figure 3 and Table 1. The basal cAMP response was almost the same as that of eel LH/CGR-wt for all mutants. In eel LH/CGR-WT, the Rmax value was 58.8 nM/10⁴ cells, showing a continuously increasing trend in a dose-dependent manner. The half-maximal effective concentration (EC_50_) was 189.7 nM/mL. In the three mutants (N49Q, N306Q, and N312Q), the EC_50_ values were 215.1 ng/mL, 233.3 ng/mL, and 235 ng/mL, respectively, indicating slightly lower cAMP responses than those in LH/CGR-wt. The Rmax values were approximately 64% and 60% of LH/CGR-wt in the N306Q and N312Q mutants, respectively, whereas the N49Q mutant showed a slight increase to 119% of LH/CGR-wt. The Rmax and EC_50_ values for the N306Q and N312Q mutants were approximately 0.8–0.81-fold that of LH/CGR-wt, suggesting that the substitution of Asn with Gln had little influence on cAMP signaling (Figure 4).

Specifically, the cAMP response in the N201Q mutant was completely impaired, even at high agonist concentrations. The N201Q mutant showed the lowest expression level (approximately 40% of the LH/CGR-wt level); however, no cAMP response was observed. Thus, we determined that, while none of the putative N-linked glycosylation sites are required for cAMP responsiveness, the specific glycosylation site at Asn201 plays a crucial role in cAMP signal transduction.

### 3.3. Loss of Cell-Surface Receptors

We optimized an ELISA method to analyze receptor surface loss following agonist treatment. Cells were preincubated with the agonist for 60 min, followed by a time-dependent analysis of receptor levels on the cell surface (Figure 5). In cells expressing eel LH/CG-wt, receptor levels were approximately 72% after the first 5 min, increasing to 83% at 15 min. The receptor loss pattern in cells expressing the N49Q mutant was similar to that in cells expressing LH/CG-wt. The surface receptor levels for the two mutants, LH/CGR-N306Q and -N312Q, were 69% and 68%, respectively, within the first 5 min. Thereafter, the level slightly increased to 82% and 80% at 15 min and gradually increased to more than 100% at 60 min. Although the N201Q mutant showed a reduction in surface receptor levels to approximately 87% in 5 min after agonist treatment, the receptors recovered to 100% by 15 min. This indicates that the decrease was less pronounced compared to other mutants, and the surface expression recovered more rapidly.

When assessing receptor loss, the reduction in cell-surface receptors at 15 min was 17% for eel LH/CGR-wt. The rate of formation of eel LH-LH/CGR complexes for the N49Q, N306Q, and N312Q mutants declined rapidly within the first 5 min. Notably, the N201Q mutant exhibited no detectable loss of the receptor from the cell surface (Figure 6). These results indicated that the N49Q, N306Q, and N312Q mutations did not significantly impact receptor loss, displaying a pattern similar to that of eel LH/CGR-wt. In contrast, the eel LH/CGR-N201Q mutant demonstrated minimal to no receptor loss at 15 min, which was likely strongly associated with its surface expression level and the lack of a cAMP response. This mutant exhibited minimal receptor loss, which was likely attributable to extremely low expression levels.

### 3.4. pERK1/2 Activation in HEK293 Cells

Next, we analyzed pERK1/2 levels in HEK-293 cells transiently transfected with each eel LH/CGR mutant plasmid. The total ERK and pERK1/2 levels were estimated by calculating the ratio of pERK1/2 to total ERK1/2 using HTRF. The basal pERK1/2 ratio was detected at a similar level in all receptors except for the N312Q mutant in agonist-pretreated cells, indicating that N312Q showed a 1.23-fold increase compared with eel LH/CGR-wt (Figure 7A). Additionally, a comparative analysis of the basal and 5 min responses of each mutant revealed that the N201Q and N312Q mutants exhibited the lowest increases, with 2.5- and 2.6-fold changes, respectively. Notably, the N312Q mutant exhibited a high basal pERK1/2 ratio, which likely contributed to the lower fold observed upon stimulation (Figure 7B).

The results, normalized to a ratio of 1 for the LH/CGR-wt at 0 min, are presented in Figure 8. pERK1/2 activity peaked at 5 min in all mutants and then gradually decreased until 30 min. The overall time-dependent pattern of pERK1/2 activity was slightly lower in the N49Q, N201Q, and N306Q mutants than in LH/CGR-wt. The N312Q mutant exhibited a pattern similar to that of LH/CGR-wt, despite a slightly higher basal response. The pERK1/2 activity of eel LH/CGR-wt showed an approximately 3.97-fold increase over the basal response 5 min after agonist treatment. The mutants (N49Q, N201Q, N306Q, and N312Q) exhibited increases of 3.3-, 2.6-, 3.1-, and 3.7-fold, respectively, compared to the basal response of LH/CGR-wt at 5 min post stimulation. We also conducted a time-dependent analysis of pERK1/2 activity using western blotting. Rapid pERK1/2 activation was detected in all mutants at 5 min, similar to the HTRF results, and this abruptly decreased by 15 min. Thus, pERK1/2 activity decreased to approximately 20% and 30% of maximum levels, respectively (Figure 9A,B). No significant differences were observed between LH/CGR-wt and the mutant proteins, and no differences were detected in the HTRF results. At 5 min, the N49Q and N201Q mutants showed a modest reduction in pERK1/2 activity compared to the wild-type. Notably, in the N201Q mutant, cAMP signaling through Gα protein was completely abolished, whereas pERK1/2 activation was preserved at normal levels. This suggests that in the N49Q and N201Q mutants, pERK1/2 activation may be mediated by alternative signaling pathways rather than the cAMP signaling pathway.

Of note, despite the complete impairment of cAMP activity, the N201Q mutant exhibited the lowest pERK1/2 response, although it was relatively high. This indicated that the two signaling pathways are clearly distinguishable based on their respective activities. Accordingly, we conclude that the cAMP and pERK1/2 pathways in the N201Q mutant exhibit pleiotropic signal transduction. Therefore, the N201Q mutant appeared to exhibit pathway-specific biased signaling with respect to cAMP and pERK1/2 activity.

## 4. Discussion

An LH/CGR is a G protein-coupled receptor with a large extracellular domain that binds with high affinity to its ligands, LH or CG. The carboxyl-terminal domain contains multiple phosphorylation sites that play roles in receptor regulation and signaling. Naturally occurring mutations have been identified in the extracellular and transmembrane domains as well as in the intracellular loop region [19]. LH/CGR has a large extracellular domain that has been shown to be essential for high-affinity ligand binding. There are potential glycosylation sites in this region [13]. In most mammals, LH/CGR contains six N-linked glycosylation sites [20,21], whereas in eels, there are four potential N-linked glycosylation sites. An additional site has been identified at amino acid position 49, encoded in exon 1. A well-conserved glycosylation site is present at amino acid position 201, encoded in exon 7, of both mammals and eels, and two additional sites are encoded in exon 10. In mammalian LH/CGR, well-conserved N-linked glycosylation sites are encoded in exons 2 and 6, and three putative glycosylation sites are encoded in exon 10. However, in eels, no corresponding glycosylation sites have been found to be encoded in exons 2 or 6, and only one site has been found to be encoded in exon 10.

Differences in the LH/CGR ratio likely arose through evolution. However, both mammals and eels have four well-conserved glycosylation sites in the FSHR. Although it is well known that rLH/CGR and hLH/CGR are glycosylated and that glycosylation is not absolutely required for hormone binding or signal transduction [2,12] it remains unknown whether the potential glycosylation sites play a significant role in signal transduction in fish species, except for recent findings on eel FSHR [16]. In this study, we determined whether N-linked glycosylation sites are necessary for eel LH/CGR-mediated signal transduction. Four mutants (N49Q, N201Q, N306Q, and N312Q) were constructed using site-directed mutagenesis to individually and collectively alter the conserved sequences of the N-linked carbohydrate sites in eel LH/CGR.

We demonstrated for the first time that the N-linked glycosylation site (Asn^201^) in eel LH/CGR regulates receptor-mediated cAMP signaling, as signaling activity was completely impaired after mutation of this site. The N201Q mutant showed the lowest cell-surface expression among the mutants, and the receptor did not undergo normal surface loss. However, we confirmed that pERK1/2 signaling proceeded normally, although at a low level. Thus, Asn^201^, among the four potential N-linked glycosylation sites encoded in exon 7, may be essential for optimal post-translation receptor stability. In addition, we reported that other mutations at other sites in eel LH/CGR (N49, N306, and N312) did not affect the PKA signal transduction pathway or pERK1/2 activity.

Our results are consistent with previously reported findings for a mutation at Asn^173^ in rLH/CGR, which is located at the same conserved position as Asn^201^ in eel LH/CGR (including the signal sequence), and the mutant protein is not expressed in the plasma membrane [8]. Additionally, the presence of functional carbohydrate chains at this site is necessary for the expression of a functional receptor on the cell surface [2,20] and is involved in the intramolecular folding of the nascent receptor [10]. Studies involving glycosidases and mutation analyses have indicated that glycan removal is not required for high-affinity receptor binding. In a study of rFSHR, the Asn^174^ mutant (encoded at the same exon 7 position as Asn^201^ in eel LH/CGR) was expressed on the cell surface and bound FSH with normal affinity, but the protein with mutations at both Asn^174^ and Asn^276^ showed no receptor binding [11]. The enzymatic treatment of wild-type FSHR typically results in high-affinity binding. However, when FSHR glycosylation is inhibited by tunicamycin B2, the receptor fails to bind FSH. Similarly, our study reported that the same glycosylation site (Asn^201^) encoded in the conserved exon 7 of eel FSHR plays a crucial role in receptor expression, cAMP signaling, and receptor surface loss. Specifically, the Asn^201^ mutation leads to only 9.2% receptor expression, an impaired cAMP response, and the absence of receptor loss from the cell surface [16]. Asn^173^ in rLH/CGR is glycosylated as demonstrated in previous studies [8,11]. Although this glycosylation site is not strictly required for high-affinity binding, it plays a significant role in cAMP enhancement. In previous studies on the rLH/CGR, among the six potential N-linked glycosylation sites, N77 was identified as a site where glycosylation does not occur, whereas N152 and N173 were reported to be glycosylated. Accordingly, it is presumed that in eel LH/CGR, glycosylation likely occurs at N201, which corresponds to N173 in rat LH/CGR. This aligns with previous reports indicating that this site plays a critical role in cAMP signaling. Therefore, the eel LH/CGR-N210Q mutant is closely involved in the signal transduction pathway, similar to the N-linked glycosylation sites located in exon 7 of rLH/CGR, hLH/CGR, and rFSHR.

Research on hTSHR and its functions has shown that the substitution of Asn at positions 177 and 198, within the consensus sequence of a potential N-linked glycosylation site, does not affect receptor binding affinity, but leads to an increase in the intracellular cAMP response [14]. Therefore, among the glycoprotein hormone receptors, the highly conserved N-linked glycosylation site at N201 encoded in exon 7 plays a crucial role in the signal transduction of LH/CGR and FSHR. These results, along with the data on eel LH/CGR-N201Q, suggest that the N-glycosylation site is critical for receptor function, particularly in the cAMP response, expression, and surface loss of receptors.

Other studies have suggested that tunicamycin treatment of protease-activated receptor (PAR) causes a marked shift in mobility and degrades mature PAR1 through continuous agonist stimulation [22,23]. In many other GPCRs, disruption of N-glycosylation perturbs receptor surface expression, as reported for PAR2, bradykinin B2 receptor, smoothened receptor, and orphan receptors [24,25,26,27]. However, certain receptors do not show detectable deficits in function when N-glycosylation is blocked, as observed for the muscarine M2 receptor, histamine H2 receptor, and the class A orphan receptor GPR61 [28,29,30]. Recent studies on N-glycan-deficient mutants have shown that the purinergic P2Y2 receptor is required for its appropriate expression and that it undergoes proteasomal degradation [31]. In group 6 members of class C GPCRs (GPRC6A), one of the seven N-glycan sites plays a crucial role in modulating receptor surface expression, whereas the others contribute to various aspects of receptor function, including signaling efficiency and stability [32]. The a1D-adrenergic receptor undergoes early termination between transmembrane domains 2 and 3 because of mutations at N65 and N82, leading to impaired membrane expression, likely as a result of degradation. This demonstrates that glycosylation is required for the complete translocation of nascent functional receptors [33]. Therefore, these N-linked glycosylation sites appear to be selectively involved in receptor expression and signaling depending on the specific glycosylation site. Furthermore, it has been confirmed that glycosylation functions differ across GPCRs, rather than being universally conserved.

Activation of pERK1/2 plays a pivotal role in mediating a wide range of physiological responses initiated by GPCR signaling [34]. The ERK1/2 pathway is activated through diverse mechanisms in response to extracellular signals, such as growth factors, hormones, cytokines, and insulin, as well as various intracellular processes and pathological conditions [35]. pERK1/2 signaling is widely recognized to be facilitated by the binding of β-arrestin to the phosphorylated terminal region of GPCRs, a process mediated by GRKs in response to ligand stimulation [36,37]. The conformational states of β-arrestin are orchestrated by ligand-induced barcoding of the receptor tail through GRKs. Consequently, biased agonists offer a remarkable opportunity to modulate signaling pathways, selectively engaging either G protein- or β-arrestin-mediated signaling [38,39,40].

In the present study, we observed that pEKR1/2 levels in eel LH/CGR peaked at 5 min, as evidenced by both HTRF and western blotting analyses. These findings align with previous reports on hCG- and FSH-induced activation, which similarly peak at approximately 5–6 min [34]. Furthermore, our study revealed that pERK1/2 activation remained robust, although slightly reduced, even in the N201Q mutant, which lacked cAMP activity. These findings are particularly significant as they align with those of previous studies on constitutively active LH/CGR, where basal cAMP levels were highly detectable even in the absence of ligand stimulation, yet pERK1/2 activation remained minimal until increasing ligand concentrations gradually enhanced phosphorylation [41]. Furthermore, LH/CGR exhibits pleiotropic effects on cAMP activation, progesterone production, IP3 signaling, and pERK1/2 activation [42].

In our study, even in cells expressing the N201Q mutant, which lacked cAMP signaling, ERK1/2 phosphorylation peaked at 5 min, increasing by more than three times the basal level. The cAMP and ERK1/2 phosphorylation signal-transduction pathways showed markedly distinct outcomes.

Based on our results, in cells expressing eel LH/CGR, pERK1/2 was activated within 5 min, similar to that observed for other GPCRs. The N201Q mutant did not undergo PKA signal transduction; however, it exhibited pERK1/2 activity, although the specific pathway remains unknown. N-linked glycosylation sites in conserved regions are not essential for pERK1/2 activation. Therefore, it can be inferred that biased signaling occurs in eel LH/CGR. The N201Q mutant can be considered an optimal model for studying biased signaling. These results are consistent with previous reports on the inactive FSHR-A189V mutant and FSHR-M512I, which show impaired G protein signaling, but not β-arrestin-dependent pERK activation, leading to intracellular retention and a decreased expression level, indicating biased signaling [43,44]. Recently, glycoprotein hormone receptors for LH/CG and FSH, as well as other GPCRs, have been reported to have biased antagonist signaling and agonism [45,46,47,48,49,50,51].

Consequently, the glycosylation sites of eel LH/CGR have been proposed to selectively regulate PKA signaling and pERK1/2 activation through biased signaling. Although specific glycosylation site (N201Q) is essential for cell-surface expression, cAMP responsiveness, and receptor surface loss, this mutation had no effect on pERK1/2 activation. In conclusion, signaling is influenced by specific GPCR or glycosylation sites. Importantly, glycosylation sites are considered one of the most crucial factors in the post-translational modification of proteins.

## 5. Conclusions

Taken together, our study demonstrated that the N-glycosylation site in eel LH/CGR-N201Q played a critical role in attenuating cAMP responsiveness and exhibited the lowest cell-surface expression level of the tested mutants, while having no effect on pERK1/2 activity. The absence of glycosylation at this site, along with its substitution with Gln, is presumed to induce conformational changes that affect post-translational modifications. Thus, the potential N201 glycosylation site, located in exon 7 of LH/CGR, may serve as a valuable model for elucidating cAMP signaling and receptor-ligand interactions in receptor-mediated complex formation. Our findings suggest that pERK1/2 activity is distinct from G-protein-mediated cAMP signaling, supporting the notion that the pERK1/2 pathway can be activated through a G protein-independent mechanism. These results provide significant insights into the role of glycosylation in regulating eel LH/CGR function and contribute to a broader understanding of GPCR signal transduction. Future studies using GPCR molecules may offer invaluable insights into the structure-function relationship of LH/CGR-LH complexes in mediating cAMP and pERK1/2 signaling. Further investigations are necessary to fully address this question. Notably, cells expressing LH/CGR-N201Q may present a novel paradigm for understanding the cellular and molecular mechanisms underlying the LH/CGR-LH complex function.

## Figures and Tables

**Figure 1 cimb-47-00345-f001:**
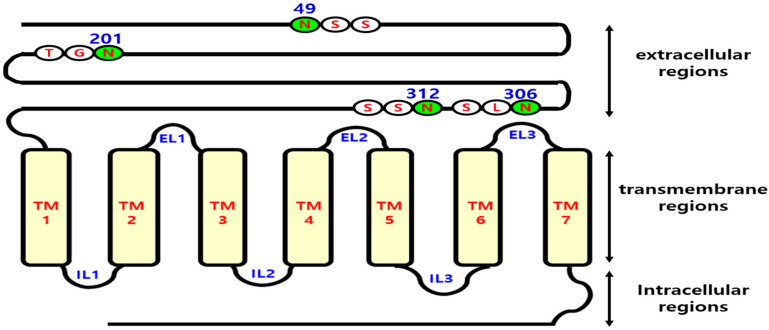
Schematic representation of the structure of wild-type eel LH/CGR and its N-linked glycosylation sites. Similar to the situation in mammals, N-linked glycosylation sites are present in the extracellular region of eel lutropin/chorionic gonadotropin receptor (LH/CGR). The extracellular domain of eel LH/CGR contains four putative N-linked glycosylation sites. Residue 49 is located in the anterior region of the N-terminus, whereas residue 201 is genetically conserved in both mammals and eels. Residues 306 and 312 are encoded in exon 10. When ligand stimulation induces phosphorylation of the C-terminal region of these receptors by GRKs, intracellular signal transduction is initiated. Therefore, signal transduction was analyzed in relation to variants at the glycosylation sites.

**Figure 2 cimb-47-00345-f002:**
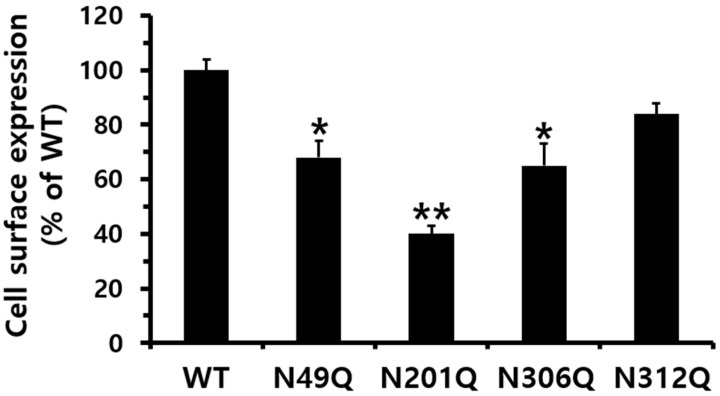
Cell-surface expression of eel LH/CGRs after transient transfection in HEK 293 cells. Wild-type eel LH/CGRs (LH/CGR-wt) and mutants were transiently expressed in HEK 293 cells, and their expression levels were determined using enzyme-linked immunosorbent assay (ELISA). Values are presented as the mean ± standard error of the mean from three independent experiments and normalized to the eel LH/CGR-wt level, which was set at 100%. Statistically significant differences were determined using one-way analysis of variance (ANOVA) followed by Tukey’s multiple comparisons test (* *p* < 0.05, ** *p* < 0.01).

**Figure 3 cimb-47-00345-f003:**
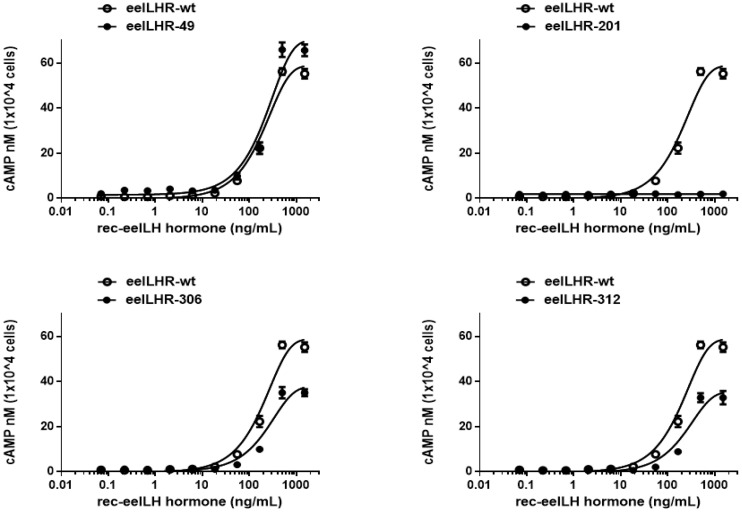
Total cAMP levels in CHO-K1 cells transfected with eel LH/CGR-wt and N-linked glycosylation site mutants. After transfection in a 6-well plate, 10,000 cells were seeded in a 384-well plate for 48 h. The cells were then incubated with rec-eel LH for 30 min, and cAMP levels were analyzed using a homogeneous time-resolved fluorescence (HTRF) assay. The ratios of the values measured at 665 and 620 nm were calculated to determine the delta F% value. Using GraphPad Prism software 6.0, these data were then compared to the cAMP standard to determine the amount of cAMP produced. Data represent the mean ± standard error of the mean of three independent experiments. Mean values were fitted to a single-phase exponential decay curve.

**Figure 4 cimb-47-00345-f004:**
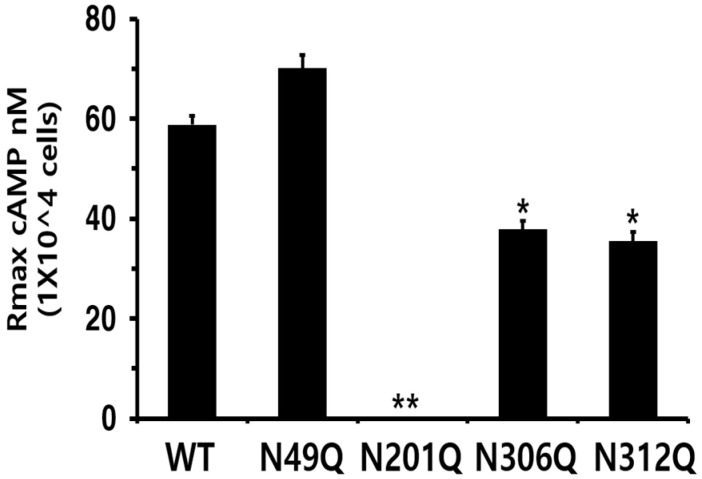
Rmax value of cAMP for eel LH/CGR-wt and N-linked glycosylation mutants. The maximum cAMP values presented in Figure 3 and Table 1 are presented as bar graphs. Statistically significant differences were determined using one-way ANOVA, followed by Tukey’s multiple comparisons test (* *p* < 0.05, ** *p* < 0.01).

**Figure 5 cimb-47-00345-f005:**
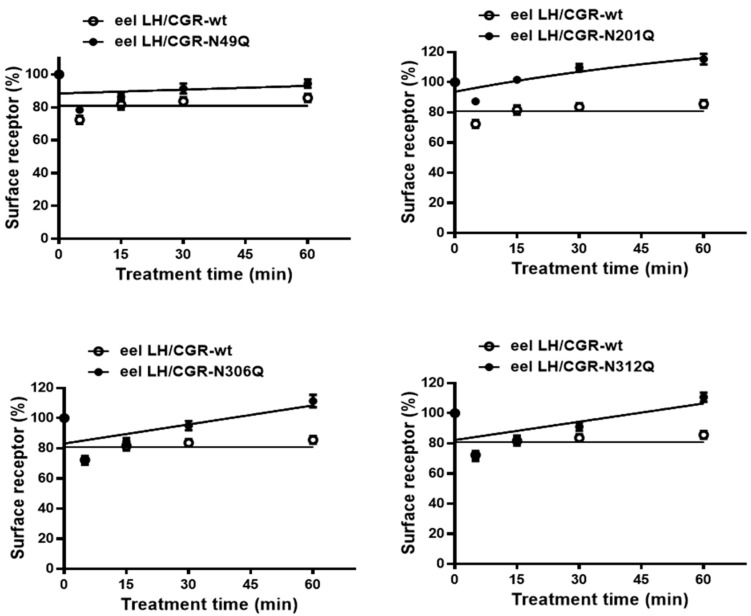
Time-dependent cell-surface loss in eel LH/CGR-wt and N-linked glycosylation mutants. The cell-surface expression level in the absence of agonist treatment was used as the baseline and set to 100% for comparison. The mean data were fitted to a single-phase exponential decay equation. The blank circles indicate the same curves for eel LH/CGR-wt.

**Figure 6 cimb-47-00345-f006:**
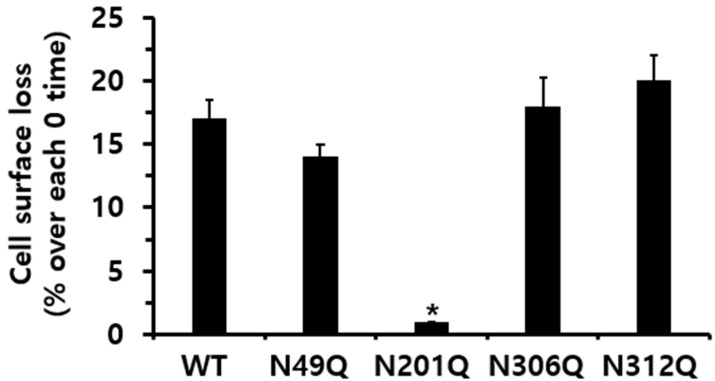
Cell-surface loss of eel LH/CGR-wt and N-linked glycosylation mutants. Receptor surface loss is expressed as a percentage, with the absence of agonist treatment considered as 0% cell-surface loss. The decrease in cell-surface expression after 15 min of recombinant LH agonist treatment is presented as a percentage. Statistically significant differences were determined using one-way ANOVA, followed by Tukey’s multiple comparisons test (* *p* < 0.01).

**Figure 7 cimb-47-00345-f007:**
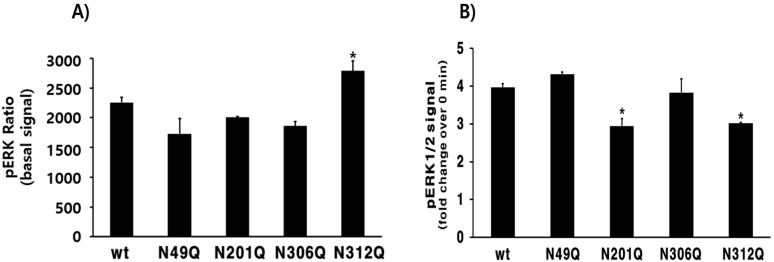
Homogeneous time-resolved fluorescence analysis of pERK1/2 activity, presenting the basal signal and the increased activity 5 min after agonist treatment. The ratio of the values measured at 665 and 620 nm was calculated, with each time-0 result presented as the basal signal. (**A**) Presentation of the basal signal of pERK1/2. (**B**) Increase at 5 min compared with the basal level for each. * *p* < 0.05, indicating a significant difference between groups.

**Figure 8 cimb-47-00345-f008:**
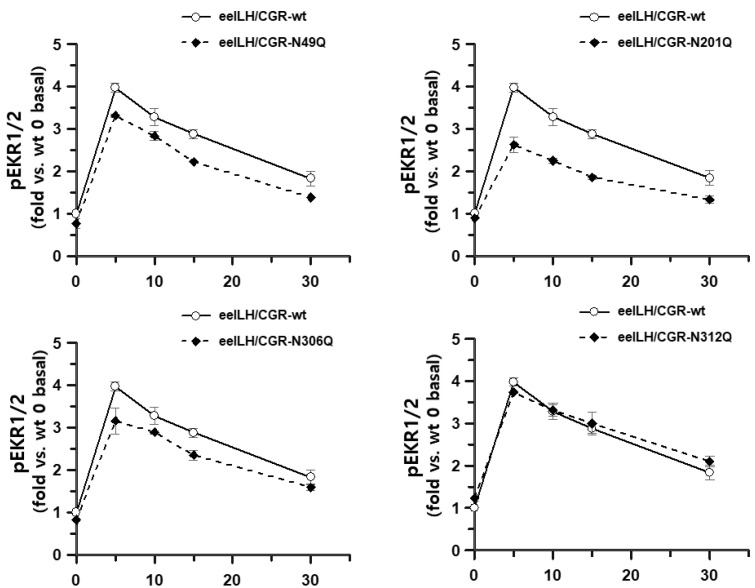
Homogeneous time-resolved fluorescence analysis results of pERK1/2 activity following treatment with recombinant eel LH for different times. All mutants were analyzed by comparing their measurements to the time-0 value before agonist treatment of eel LH/CGR-wt. The fold change values were calculated by setting the eel LH/CGR-wt time-0 value to 1, and they are presented as a graph. All mutants and LH/CGR-wt showed a sharp increase, reaching a peak value at 5 min, followed by a gradual decrease.

**Figure 9 cimb-47-00345-f009:**
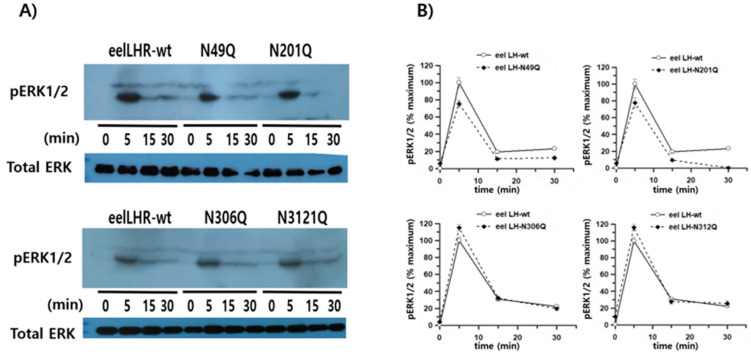
HpERK1/2 activation in HEK 293 cells transfected with eel LH/CGR-wt and mutants after stimulation with recombinant eel LH agonist. After transfection of HEK 293 cells with each plasmid, the cells were serum-starved for approximately 6 h before stimulation with 250 ng/mL agonist for the indicated time periods. Whole-cell lysates were isolated and analyzed for pERK1/2 and total ERK levels. (**A**) Western blotting results for pERK1/2 and total ERK levels. (**B**) Quantified pERK1/2 levels normalized to total ERK levels are expressed as a percentage of the maximal response, which was set to 100% for eel LH/CGR-wt at 5 min. Densitometry was performed to quantify the pERK1/2 bands.

**Table 1 cimb-47-00345-t001:** Bioactivity of eel LH/CG receptors in cells expressing glycosylation site mutant receptors.

eel LH/CG Receptors	cAMP Responses		
	Basal ^a^ (nM/10^4^ Cells)	EC_50_ (ng/mL)	Rmax ^b^ (nM/10^4^ Cells)
LH/CGR-wt	0.1 ± 0.1	189.7 (1.0-fold) (159.1 to 235.0) ^c^	58.8 ± 1.8 (100%)
LH/CGR-N49Q	1.6 ± 0.9	215.1 (0.88-fold) (173.2 to 283.7)	70.1 ± 2.7 (119%)
LH/CGR-N201Q	- ^d^	-	-
LH/CGR-N306Q	0.1 ± 0.1	233.3 (0.81-fold) (182.2 to 324.1)	37.8 ± 1.8 (64%)
LH/CGR-N312Q	0.1 ± 0.2	235.0 (0.80-fold) (181.9 to 331.9)	35.5 ± 1.8 (60%)

Values are the means ± SEM of triplicate experiments. Log (EC_50_) values were determined from the concentration–response curves obtained from in vitro bioassays. ^a^ Basal cAMP level average without agonist treatment. ^b^ Rmax average cAMP level/104 cells. ^c^ Geometric mean (95% confidence limit) of at least three experiments. ^d^ Nondetectable.

## Data Availability

All relevant data are contained within the article.

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
