# Peer review of "The N-Linked Glycosylation Site N201 in eel Lutropin/Choriogonadotropin Receptor Is Uniquely Indispensable for cAMP Responsiveness and Receptor Surface Loss, but Not pERK1/2 Activity"

_cimb, 2025, doi:10.3390/cimb47050345_

Round 1

Reviewer 1 Report

Comments and Suggestions for Authors

In this manuscript, the authors determined the role of N-linked glycosylation sites for eel LH/CGR cell surface expression and its internalisation, causing ERK1/2 phosphorylation and inducing cAMP accumulation in the agonist-stimulated cells. For this purpose, four mutants (N49Q, N201Q, N306Q, and N312Q) were created using site-directed mutagenesis. The results presented in this manuscript demonstrate that the N-glycosylation site at N201 is important for the receptor cell surface expression, internalisation, potentiation of cAMP production, but not for the ERK1/2 phosphorylation. Based on the results presented in the manuscript, the author suggested that the potential N201 glycosylation site in eel LH/CGR is important for the receptor’s biased signalling. Although this manuscript contains some valuable scientific data, it can be improved by addressing the following comments.

  1. Please explain in detail how the WT and mutants were sub-cloned into the expression vectors (see line 108).
  2. In Figure 5, the N201Q mutant also showed a reduction in the receptor levels at the cell surface within the first 5 min of against-stimulation (see line 286). Please explain.
  3. Did the N201Q mutant show reduced pERK1/2 activity even after normalising it to cell surface levels of the receptor (see Figure 7B)?
  4. No difference between the N49Q and N201Q mutants in pERK1/2 activation at 5 min (see Figure 9B). Please explain.

In addition, there are a few typos and grammatical errors which need to be corrected (for example, change ‘Ga’ to ‘Gα’ [lines 17 and 46], add ‘full stop’ after reagent [line 132], change ‘Asn’ and ‘Gln’ to ‘Asn (N)’ and ‘Gln (Q)’ [lines 207-208], change ‘the glycosylation mutants’ to ‘mutation of the glycosylation sites’ [line 227] and change ‘sites (N201Q) are’ to ‘site (N201Q is’ [line 481]).

Author Response

Reviewer 1

Comments and Suggestions for Authors

In this manuscript, the authors determined the role of N-linked glycosylation sites for eel LH/CGR cell surface expression and its internalisation, causing ERK1/2 phosphorylation and inducing cAMP accumulation in the agonist-stimulated cells. For this purpose, four mutants (N49Q, N201Q, N306Q, and N312Q) were created using site-directed mutagenesis. The results presented in this manuscript demonstrate that the N-glycosylation site at N201 is important for the receptor cell surface expression, internalisation, potentiation of cAMP production, but not for the ERK1/2 phosphorylation. Based on the results presented in the manuscript, the author suggested that the potential N201 glycosylation site in eel LH/CGR is important for the receptor’s biased signalling. Although this manuscript contains some valuable scientific data, it can be improved by addressing the following comments.

  1. Please explain in detail how the WT and mutants were sub-cloned into the expression vectors (see line 108).

→The receptor cDNA cloned into pGEM-T easy vector was digested with XhoI and EcoRI restriction enzymes, whose recognition sites were introduced by PCR primers. The digested fragments were then inserted into the corresponding sites of the pVSV-G expression vector. The full-length eel LH/CGR cDNA, including signal sequence, was also digested with EcoRI and XhoI, and subcloned into the same sites of pcDNA3 mammalian vector” in Line 107-114.

  1. In Figure 5, the N201Q mutant also showed a reduction in the receptor levels at the cell surface within the first 5 min of against-stimulation (see line 286). Please explain.

→We explained that “Although the N201Q mutant showed a reduction in surface receptor levels to approximately 87% in 5 minutes after agonist treatment, the receptors recovered to 100% by 15 minutes. This indicates that the decrease was less pronounced compared to other mutants, and the surface expression recovered more rapidly” in Line 286-289.

  1. Did the N201Q mutant show reduced pERK1/2 activity even after normalising it to cell surface levels of the receptor (see Figure 7B)?

→Figure 7 B indicates to increase in 5 min compared with the basal level for each. Thus, it is not a cell surface level but an increased level with the basal level in N201Q mutants. Thus, we suggest that “N201Q and N312Q mutants exhibited the lowest increases, with 2.5- and 2.6-fold changes, respectively” in Line 319-320.

  1. No difference between the N49Q and N201Q mutants in pERK1/2 activation at 5 min (see Figure 9B). Please explain.

We inserted “ At 5 minutes, the N49Q and N201Q mutants showed a modest reduction in pERK1/2 activity compared to the wild-type. Notably, in the N201Q mutant, cAMP signaling through Gα protein was completely abolished, whereas pERK1/2 activation was preserved at normal levels. This suggests that in the N49Q and N201Q mutants, pERK1/2 activation may be mediated by alternative signaling pathways rather than the cAMP signaling pathway” in Line 345-350.

In addition, there are a few typos and grammatical errors which need to be corrected (for example, change ‘Ga’ to ‘Gα’ [lines 17 and 46], add ‘full stop’ after reagent [line 132], change ‘Asn’ and ‘Gln’ to ‘Asn (N)’ and ‘Gln (Q)’ [lines 207-208], change ‘the glycosylation mutants’ to ‘mutation of the glycosylation sites’ [line 227] and change ‘sites (N201Q) are’ to ‘site (N201Q is’ [line 481]).

→We all changed by reviewer’s comments. But we could not confirm the “add “full stop” after reagent [Line 132].

Reviewer 2 Report

Comments and Suggestions for Authors

The authors have studied the eel LH receptor, and the effect of the glycosylation in the extracellular domain on the function of the receptor.
It is interesting to note that receptors with mutations in N201, which have been identified in many species, showed characteristic functional changes.
Since there are parts of the paper where additions and modifications are desirable for consideration, the following comments should be taken into account.
In particular, it is important to confirm whether the receptor actually binds to the ligand or not.

-If there is any previous report on the N201 mutant receptor that discusses its function, please discuss how it relates to the present results.

-Figure 2: N201Q shows predominantly lower receptor expression, could this be due to differences in transfection efficiency?

-Figure 2: The amount of receptor expressed is shown here, but it would be desirable to quantify the amount of ligand that binds to the receptor.

-Figure 3: Here, cAMP production is expressed as a per-cell amount, but if the function of the receptor is to be evaluated, it would be more desirable to express the amount produced per binding hormone.

-Figure 5: Here, the disappearance of the receptor from the cell surface is measured. Why is a 60-minute pre-incubation with agonist added? Wouldn't sequestration of receptors already occur during this time?

-Is it correct to think that the changes observed in N201Q were caused solely by the presence or absence of glycosylation? Is it possible that the conformation of the receptor as a whole has changed as a result of the mutation?

Author Response

Reviewer II

Comments and Suggestions for Authors

The authors have studied the eel LH receptor, and the effect of the glycosylation in the extracellular domain on the function of the receptor. It is interesting to note that receptors with mutations in N201, which have been identified in many species, showed characteristic functional changes. Since there are parts of the paper where additions and modifications are desirable for consideration, the following comments should be taken into account. In particular, it is important to confirm whether the receptor actually binds to the ligand or not.

-If there is any previous report on the N201 mutant receptor that discusses its function, please discuss how it relates to the present results.

→We discussed these results from Line 411 to Line 429. In this sentence, we suggested that rLH/CGR (Asn173: the same site at the eel LH/CGR in which does not include the signal sequence), hLH/CGR, rFSHR, and hTSHR are located to the conserved regions at the exon 7, demonstrating that these sites are related to the receptor expression, and impaired cAMP response. We also introduced these contents of rLH/CGR at Line 67 to 70. Thus, we inserted “In previous studies on the rat, among the six potential N-linked glycosylation sites, N77 was identified as a site where glycosylation does not occur, whereas N152 and N173 were reported to be glycosylated. Accordingly, it is presumed that in eel LH/CGR, glycosylation likely occurs at N201, which corresponds to N173 in rat LH/CGR. This aligns with previous reports indicating that this site plays a critical role in cAMP signaling. Therefore, the eel LH/CGR-N210Q mutant is closely involved in the signal transduction pathway, similar to the N-linked glycosylation sites located in exon 7 of rLH/CGR, hLH/CGR, and rFSHR “at the Line 429-436.

-Figure 2: N201Q shows predominantly lower receptor expression, could this be due to differences in transfection efficiency?

→We tested many times to determine the loss of receptor expression. Thus, the transfection efficiency does not matter. N201Q mutant was only 40% of that of LH/CGR-wt, showing that it has an extraordinary influence on cell surface expression among the glycosylation sites.

-Figure 2: The amount of receptor expressed is shown here, but it would be desirable to quantify the amount of ligand that binds to the receptor.

→We agree with the reviewer’s comments. However, we did not analyze the receptor binding assay. And the unsubmitted results for the equine LH/CGR are almost identical to eel LH/CGR in the receptor expression and cAMP response.

-Figure 3: Here, cAMP production is expressed as a per-cell amount, but if the function of the receptor is to be evaluated, it would be more desirable to express the amount produced per binding hormone.

→ In this cAMP analysis, we evaluated total cAMP response at the 96-well plates. Of course, if we tested the receptor binding assay, it will be addressed to the per ligand binding. In this paper, we suggest cAMP level by dose-dependent manner.

-Figure 5: Here, the disappearance of the receptor from the cell surface is measured. Why is a 60-minute pre-incubation with agonist added? Wouldn't sequestration of receptors already occur during this time?

→In surface loss experiment, the receptors normally internalized and partly degradation at the lysosome and in part recycled from endosome to cell surface. Of course, receptors may be in part degradation during this reaction. But we confirmed that the culture time for 60 min does not affect the recycling of the receptor. Thus, we reacted to 60 min at the pre-incubation time.

-Is it correct to think that the changes observed in N201Q were caused solely by the presence or absence of glycosylation? Is it possible that the conformation of the receptor as a whole has changed as a result of the mutation?

→Of course, Asn152 and Asn173 (N201 in eel LH/CGR including signal sequence) in rLH/CGR is glycosylated as demonstrated in previous studies (Ref 8 and 11). The other sites (Asn77) was not glycosylated. Thus, we stated “The absence of glycosylation at this site, along with its substitution with Gln, is presumed to induce conformational changes that affect post-translational modifications “ at Line 513 to 514 of the conclusion section.

Round 2

Reviewer 2 Report

Comments and Suggestions for Authors

Thank you for the authors' appropriate responses to the reviewers' comments. Given that this is a study dealing with receptors, presenting the results of binding experiments with ligands would further enhance the completeness of the paper. However, I believe the current paper still presents results worthy of publication.